# Factors Associated with Older People’s Anxiety Symptom Positioning after COVID-19: Cross-Sectional Findings from a Canadian Sample

**DOI:** 10.3390/healthcare12181837

**Published:** 2024-09-13

**Authors:** Gail Low, Anila Naz AliSher, Juceli Morero, Zhiwei Gao, Gloria Gutman, Alex Franca, Sofia von Humboldt

**Affiliations:** 1Faculty of Nursing, University of Alberta, Edmonton, AB T6G 1C9, Canada; 2College of Nursing, University of Sao Paulo at Ribeirão Preto, Sao Paulo 14040-902, Brazil; morero.juceli@gmail.com; 3Faculty of Medicine, Memorial University, St. John’s, NL A1B 3V6, Canada; zgao@mun.ca; 4Gerontology Research Centre, Simon Fraser University, Vancouver, BC V6B 5K3, Canada; gutman@sfu.ca; 5Laboratory of Human Development and Cognition, Federal University of São Carlos, São Paulo 13566-590, Brazil; alex.tonante@gmail.com; 6William James Center for Research, ISPA—Instituto Universitário, 1149-041 Lisbon, Portugal; sofia.humboldt@gmail.com

**Keywords:** anxiety, coping behaviors, older adults, survey, logistic regression

## Abstract

This study sheds light on the personal characteristics of older Canadians self-identifying as severely anxious and the coping strategies that they gravitated to mitigate their anxiety. Our studied sample consisted of 606 Canadians aged 60 and above who took part in an e-survey across all 10 of Canada’s provinces, launched in July 2022, when social distancing was lifted across the country. Participants completed a personal characteristics questionnaire, the Geriatric Anxiety Scale or GAS-10, and a checklist of everyday coping strategies for mitigating anxiety. A seemingly greater number of severely anxious Canadians were born female, self-identified as a cisgender woman, and were in their 60s and in poor to fair health. A univariate logistic regression analysis revealed that all such personal characteristics were associated with statistically significantly greater odds of experiencing severe anxiety. In our multivariate logistic regression analysis, no significant differences were observed between the sexes (*AOR* = 0.590, *p* = 0.404), and non-binary and cisgender men (*AOR* = 0.689, *p* = 0.441) and women (*AOR* = 0.657, *p* = 0.397). Nor were there statistically significant differences in the odds of experiencing severe anxiety for those living with versus without a life partner and chronic illnesses. Older Canadians experiencing severe anxiety were far more likely to normalize their fear and anxiety (*AOR* = 4.76, *p* < 0.001), challenge their worries (*AOR* = 5.21, *p* < 0.001), and to relax or meditate (*AOR* = 2.36, *p* = < 0.001). They were less inclined to decrease other sources of stress in their lives, to stay active, and to get enough sleep. We offer anticipatory guidance for mental health program planners and practitioners, and fruitful avenues of inquiry for researchers.

## 1. Introduction

Aging has been defined as a biological and a social construct shaped by cultural norms around what constitutes ‘old’ in terms of people’s characteristics and contributions to society at large [1]. In the early stages of COVID-19, ‘old’ centered around illness and death [2,3], and social isolation [4] and loneliness [5]. While older people were characterized as an anxiety-riddled demographic, anxiety can function like a natural alarm system, motivating people to take remedial action [6]. Older people were found to be resourceful and to use a wide array of mental, behavioral, and emotional coping strategies at different points in the pandemic to manage sources of mental stress [7], and stress and fear related to COVID-19 [8]. Advancing age was deemed a protective factor in large-scale observational studies in Canada [9,10] and abroad, for example, see [11]. Other older people spoke of silver linings [12] and upsides [13] to social lockdowns.

Anxiety characterized by excessive fear, worry, and behavioral disturbances can impair everyday functioning [14]. Very early in the COVID-19 pandemic, 11% of older Canadians reported experiencing moderate to severe symptoms of anxiety [15]. During the first year of this pandemic, nearly 2 in 10 were experiencing difficulties mitigating any perceived anxiety [16]. While these findings suggest, as others [17,18,19,20] have, that older people tend to be more anxiety-resistant than anxiety-prone, when social distancing lifted, older Canadians were as prone to self-identify with severe anxiety and with no anxiety at all [21].

The ‘who’ and the ‘how’ of severe anxiety warrants greater attention to help mental health practitioners and program planners working with older people. Nationally representative evidence about who is most at risk and what lessens and enhances this risk is essential evidence for practitioners and program planners [22], particularly after COVID–19 [23]. There is also tremendous stigma attached to mental health help seeking [24].

In this context, we analyzed data collected in a nationally representative study about how older Canadians were self-managing pandemic-related anxiety [21] and social isolation [25] across all 10 provinces. Older people are experts on their position in life, including their health and what helps and hinders it [26], and our best educators. We undertook a secondary analysis of these data to explore (a) the personal characteristics of older Canadians who self-identified with severe anxiety, and (b) the everyday coping strategies that they were more and/or less inclined to gravitate to.

## 2. Methods

### 2.1. Study Design, Population, and Sample

This study is a secondary analysis of an online nation-wide survey previously conducted by our team [21] to study self-managing pandemic-related anxiety and social isolation. This study took place when social distancing had lifted across Canada. Inclusion criteria were being 60 years of age and older, residing in Canada, and being an active Qualtrics survey research panel member. Data were collected using the Qualtrics e-survey platform [27]. Qualtrics collected data across 10 Canadian provinces using census-based sampling [28]. We sought a studied sample mirroring the characteristics of Canadians at large in terms of their age, sex, and education. When our survey launched, Canadians 60 years of age and older tended to be most at risk for being hospitalized and needing intensive care for COVID-19 infections and death [29]. The e-survey included questions about anxiety, everyday coping strategies, and personal characteristics, including health. The e-survey opened on 1 July 2022 and closed on 15 August 2022, when older Canadians stopped responding altogether. At that point in time, *N* = 1327 older Canadians had completed the study e-survey.

### 2.2. Measure and Instruments

#### 2.2.1. Outcome Variable

##### Anxiety

The outcome variable was anxiety, which was measured using the Geriatric Anxiety Scale (GAS-10), which had a robust internal consistency reliability (*α* = 0.92). This lay-friendly tool includes items such as ‘I had a hard time sitting still’ and ‘I felt like something terrible was going to happen to me’. All items are rated on a scale from 0 (not at all) to 3 (all of the time). Total scores can range from 0 to 30. The clinical cut-off criterion for severe anxiety is a GAS-10 score of 12 or higher [30,31]. This publicly accessible scale can be used free of charge, provided that researchers reference it: https://gerocentral.org/wp-content/uploads/2013/03/GAS-10-item-version-2015-1-15.pdf (accessed on 1 August 2022). 

#### 2.2.2. Covariables

##### Personal Characteristics

Participants were asked about their age (60–69, 70–79, 80 years of age and older), sex at birth (male, female), gender identity (transgender, cisgender woman, cisgender man), educational status (no post-secondary education, post-secondary education), and marital status (with a life partner, without a life partner). Participants were also asked about their overall health (poor/fair, good, very good/excellent) and whether they had any chronic illnesses (no chronic illness, one chronic illness, and two or more chronic illnesses).

##### Coping Strategies

We asked participants about everyday coping strategies to manage their anxiety using the Centre for Addiction and Mental Health personal checklist [32] (α = 0.76). Older Canadians responded ‘Yes’ or ‘No’ to using any of one of 16 strategies like staying active, challenging anxious thoughts and worries, and tuning into but knowing when to take a break from news about COVID-19. This checklist remains available for public consumption§; https://www.camh.ca/en/health-info/mental-health-and-covid-19/coping-with-stress-and-anxiety (accessed on 7 September 2024).

### 2.3. Statistical Analyses

The focus of this study is on two groups from our nationally representative studied sample (*N* = 1327), older Canadians who self-identified as having ‘no anxiety at all’ and therefore had a GAS-10 score of 0 (*n* = 290; 20.6%). The other group consisted of ‘severely anxious’ participants self-identifying with a GAS-10 score of ≥12 (*n* = 316; 21.1%). We identified the dependent variable in this study as the presence of severe anxiety (GAS-10 ≥ 12). Older Canadians who self-identified as ‘severely anxious’ are likely to be the demographic that mental health practitioners and program developers are acutely interested in. Accordingly, we used descriptive statistics to summarize the personal characteristics that make severely anxious Canadians stand out from their non-anxious counterparts. Hence, we used frequency counts (n) and percentages (%). Significant differences in personal characteristics were examined by a chi-squared test (χ^2^) and their magnitude using the Phi coefficient (φ) [33].

In a binary logistic regression, older Canadians who self-identified were coded as ‘1’ because we wanted to estimate the probability or likelihood that severe anxiety was present if a participant was, for example, of a certain age or level of education, or used a certain coping strategy. We could identify personal characteristics and coping strategies statistically significantly increasing and/or decreasing the likelihood of being ‘severely anxious’. A final adjusted multivariate logistic regression model consisted of age, sex at birth, gender identity, number of chronic illnesses, perceived health, education, and marital status. Goodness of fit was determined using the Hosmer–Lemeshow chi-square statistic. All statistical analyses were performed with SPSS Version 29.0 (IBM Corporation, Armonk, NY, USA). 

Our sample size was also adequate for a multivariate logistic regression analysis, with 7 independent variables with 18 categories of personal characteristics and 16 coping behaviors; G*Power software Version 3.1 revealed a minimum sample size of *n* = 153 per group for detecting significant associations (α = 0.05; power = 0.80) [34]. Because we know very little about how older people were self-managing their anxiety after COVID-19, we were equally interested in personal characteristics and coping strategies that were not statistically significant. Any such tenuous personal characteristics and coping strategies were characteristics and strategies that could lessen and/or heighten the probability of older Canadians at large being ‘severely anxious’. This way, mental health practitioners and program planners stand to gain further insight into the complexities of anxiety-related recovery work at a key turning point in the COVID-19 pandemic.

### 2.4. Ethical Implications

Ethical approval (Pro0092157) was granted by the University of Alberta’s Human Ethics Review Board. This meant that all participants provided informed consent before participating in the survey (Appendix A). Qualtrics collected our data to ensure that confidentiality and anonymity of the participants was strictly maintained throughout the study, all of whom were assigned a one-time responder-specific identifier. The research team also had no access whatsoever to participants’ names, IP addresses, or emails, ensuring their anonymity during recruitment. Accordingly, on 16 August 2022, we received a scrubbed dataset containing completely anonymized data collected from *N* = 1327 older Canadians. All such anonymous data were housed on an encrypted computer in a secured office.

Mental health matters should never be taken lightly. We wanted to make prudent recommendations to mental health practitioners and program planners. We therefore adopted a more stringent *p*-value criterion (*p* < 0.01) to establish statistical significance and generated bootstrapped 95% confidence intervals for a more plausible range of odds that certain personal characteristics and coping strategies are more and/or less associated with experiencing severe levels of anxiety.

## 3. Results

As shown in Table 1, statistically significantly greater numbers of severely anxious participants were born as female (*n* = 196; 63.4%) and identified as such (*n* = 149; 52.7%), and tended to be in their 60s (*n* = 211; 67%) and in poor/fair self-rated health (51.1% versus 15.3% among those with no anxiety at all). Observed differences based on participant age, sex, and gender were small, and for health, these were large.

A crude binary logistic regression analysis revealed that age, sex at birth, gender identity, and perceived health were statistically significantly associated with severe anxiety. Education level, marital status, and being chronically ill were not. The unadjusted odds ratios with confidence intervals are shown in Table 2.

In the adjusted logistic regression analysis (Table 3), the variables with statistically significant *p*-values in the crude logistic analysis were adjusted by each other, and for additional covariates, including gender identity, number of chronic illnesses, education, and marital status. For practical purposes, we were interested in both.

With respect to personal characteristics, being between 70 and 79 years of age (AOR = 0.437, 95% CI 0.248–0.772, *p* = 0.004) and 80 years of age and older (AOR = 0.230, 95% CI 0.098–0.543, *p* < 0.001) were associated with a lower chance of experiencing severe anxiety. Being in good (AOR = 0.196, 95% CI 0.110–0.351, *p* < 0.001) and very good to excellent (AOR = 0.138, 95% CI 0.068–0.279, *p* < 0.001) health was associated with a lower chance of experiencing severe anxiety. 

Other findings suggest that participants born male (AOR = 0.59, 95% CI 0.171–2.036, *p* = 0.404) and self-identifying as male (cisgender woman: AOR = 0.689, 95% CI 0.267–1.777, *p* = 0.441; cisgender man: AOR = 0.657, 95% CI 0.249–1.734, *p* = 0.397) had a lesser chance of experiencing severe anxiety, but not statistically significantly so. So too did chronically ill participants (one chronic illness: AOR = 0.785, 95% CI 0.531–1.189, *p* = 0.264; two or more chronic illnesses: AOR = 0.897, 95% CI 0.614–1.310, *p* = 0.573). Other tenuous personal characteristics were having a post-secondary education (AOR = 1.377, 95% CI 0.811–2.324, *p* = 0.239) and a life partner (AOR = 1.026, 95% CI 0.620–1.697, *p* = 0.920).

In keeping with our second aim, we identified coping behaviors that severely anxious participants were more and/or less inclined to gravitate to. Strategies that they were more inclined to try were accepting that some fear and anxiety were normal (AOR = 4.76, 95% CI 2.031–11.200, *p* < 0.001), challenging worries and anxious thoughts (AOR = 5.21, 95% CI 2.663–10.228, *p* < 0.001), and practicing relaxation and meditation (AOR = 2.36, 95% CI 1.365–4.112, *p* = 0.002). Older Canadians at large who accept and/or challenge felt anxiety could be 10 to 11 times more likely to experience severe anxiety. Severely anxious participants were far less inclined to decrease other sources of stress (AOR = 0.39, 95% CI 0.211–0.729, *p* = 0.003), to stay active (AOR = 0.21, 95% CI 0.110–0.408, *p* < 0.001), and to get proper rest and sleep (AOR = 0.52, 95% CI 0.275–0.995, *p* = 0.048).

Our findings revealed a number of tenuous strategies (Table 3). Severely anxious participants were generally somewhat less likely to seek credible information, engage in structured problem-solving, remember that they were resilient, be kind to themselves, and eat healthily. They were also generally somewhat more likely to have balanced media consumption, unplug from electronic devices, seek out loved ones, avoid substance use, and watch their caffeine intake. All associated confidence intervals suggest that trying any such strategies could enhance or lessen one’s prospects of being severely anxious. For example, seeking credible information might afford older Canadians at large a 50.5% lesser likelihood or up to a 54% greater likelihood of experiencing severe anxiety. If older Canadians in the general population remembered that they were resilient (OR = 0.54, 95% CI 0.267–1.092, *p* = 0.086) and dealt with problems in a structured way (OR = 0.62, 95% CI 0.310–1.249, *p* = 0.182), they were generally less inclined to be severely anxiety. Use of the latter two strategies could also tip their anxiety scale into the severity zone.

## 4. Discussion

The COVID-19 pandemic is a historic global event enhancing the risk for acquiring a life-threatening illness and death, particularly for older persons within the first two years [29,35]. Pandemic-related anxiety is on the rise globally [14], as is living in isolation from others [36]. Research studies that offer healthcare practitioners and program planners a nationally representative snapshot of older persons seemingly more prone to severe anxiety and self-selected strategies helping and perhaps even hindering them is important and timely. We analyzed data collected in our parent study across 10 Canadian provinces to compare and contrast (a) the personal characteristics of older Canadians self-identifying with severe to very severe anxiety, and (b) the everyday coping strategies that they are more and/or less inclined to gravitate to.

### 4.1. Personal Characteristics

Personal characteristics can offer anticipatory guidance for practitioners and program developers as to the who’s who of meddlesome anxiety post-COVID-19. A higher percentage of severely anxious participants were 60–69 years of age, which is consistent with previous literature elsewhere [37,38,39]. We found, as others have in the first year of the pandemic [9,40], that being in one’s 80s and 90s is an asset. Consistent age-specific patterns of findings at different time points in the pandemic lend further credence to age-anticipatory clinical assessment. This is especially important given that Canadians in their 80s can expect to have a mere 1.3 additional life years left to live [41]. Longitudinal studies are necessary to best understand age-related trajectories of mentally healthy recovery work after COVID-19, however. Whether any of these age-specific patterns fall out differently later on remains to be seen.

Perceived health was also another between-anxiety-group differentiator. Literature reviews prior to our study tell of being in very good to excellent perceived health as a mental health asset in later life [42]. Feeling less health-related vulnerability has been associated with remarkably greater tolerance for uncertainty during COVID-19 [43]. Perhaps older people in better health feel more protected against getting or recovering from COVID-19, or are more forgiving in the anticipation thereof. Experiencing better mental well-being over time appears to be associated with self-compassion [44]. Health-related mindsets may become even more important long after COVID-19, given that our years lived in good health are expected to come to a close as early as age 71 [41]. A follow-up qualitative research study will permit exploring with depth the relationships between health mindsets and expectancies.

The number of chronic illnesses was found not to be a statistically significant between-anxiety-group differentiator. In earlier studies, chronically ill older Canadians reported significantly higher symptoms of depression [10] but not anxiety [45]. Findings about chronic illnesses have tended to be more compelling for depression than for anxiety in others’ studies abroad [46,47] as well. In this study, the OR for this independent variable suggests that older Canadians living with chronic illnesses have somewhat lower odds of severe anxiety but not significantly so. We therefore wondered whether older people, presumably having to adjust their home, work, and social lives to best manage a chronic illness, are better able to take actual or potential COVID-19 health-related fallouts in their stride. Older people living with multiple chronic illnesses may be even more adept negotiators. Perhaps having to eke out a livelihood for oneself whilst chronically ill brings heightened self-compassion [44] or an ability to genuinely see that there are no guarantees in life and to not fixate on own-health hardships [44,48]. Older Canadians better able to persevere alongside the unexpected and longstanding intrusiveness of multiple chronic illnesses appear to express less COVID-19 worry [18].

Groups living at risk for infectious illnesses are well aware of how to best navigate their surrounding environments to stay as healthy and as well as possible [49]. Early in the pandemic, Gutman et al. [50] found that chronically ill older people were more in favor of masking than their midlife counterparts. Throughout COVID-19, along with highly anxious and older Canadians, who might also self-identify as a likely candidate for getting COVID-19, they were characterized as public health measure followers [51]. Other older people in their 70s and 80s have been more inclined to get tested for COVID-19 [52]. When social distancing lifted, older people across Canada, most of whom were chronically ill, reminded contemporaries to stay vigilant about their physical surroundings and, to a lesser extent, public health measures [25]. At this time, people in their 60s and older across Canada were also less prone than middle-aged and young adults to test positive for COVID-19 [53]. As of this July, some 391,000 Canadians in their 60s (versus around 755,000 people in their 20s) had been infected with the COVID-19 virus.

In the first two years of the pandemic, in Canadian [10,54] and international studies [38,55,56], older men were a consistently significantly anxiety-advantaged demographic. Some blame this on the increased burden that women experience with caregiving responsibilities [18,57] or heightened potential exposure to domestic violence in the wake of pandemic-related stresses and strains [40,56]. A recent meta-analysis also revealed that at any-age men tend to express fewer fears and anxieties about the pandemic, with this perhaps owing to social norms about standing on one’s own two feet and keeping one’s emotions in check [58]. In our study, men were generally but not always less likely to experience severe anxiety. Chima et al. [59] found that cisgender men were significantly less severity-inclined than cisgender women and even more so compared to non-binary or transgendered persons. Others’ research also tells us that older people self-identifying as LGBT were significantly more prone to anxiety and to a host of COVID-19 stressors, including access to mental health care and financial strain [60], and to more depression and loneliness [61] throughout their 60s, 70s, and 80s than same-age heterosexual persons were. Given the tenuous nature of our findings in relation to sex and gender identity, objective assessments with mixed expectancy and open-ended questions are prudent.

Our findings suggest that people with severe anxiety are not significantly more likely to be without a life partner or to have sought support from a loved one, and this surprised us. In the first year of the pandemic, living in isolation from others was detrimental to older Canadians’ perceived mental health [62,63]. Older Canadians also reported being far less prone to report familial conflicts than midlife adults [50]. In the first two years of the pandemic, news about older people’s heightened risk for COVID-19-related harms became part of the fabric of everyday life [64]. We are now coming to learn, as others forewarned [65], of the anxiety-provoking effects of negative news [66], any television news [67], and rehashed news among online acquaintances [68] about COVID-19. We therefore wondered whether participants’ support seeking might be seen as exacerbating loved ones’ worries about them. Perhaps not having a partner also living at a heightened risk for COVID-19 offsets this worry. We should not assume that older people with a life partner are far better off in terms of their mental health.

In keeping with others’ findings about older Canadians [40,45], the same can be said with respect to education, wherein more is not better. Perhaps it is not book wisdom but rather, as Ardelt and Jeste found, life wisdom that helps older people be open to experiences and to expect to grow in some way when experiencing crises in their life [48]. Expressing discomfort around not knowing what the future will bring [43] and ruminating over what is yet to come [69] can be a mentally detrimental and a lonely headspace to be in. Open- and present-mindedness seems to be a salutary approach [70].

### 4.2. Coping Strategies

A global pandemic is hardly an everyday experience, particularly when transitioning from one’s own four walls into open spaces with COVID-19 still lingering. There is a good deal of literature suggesting that confronting reality and accepting one’s circumstances can diminish fear and anxiety [35,71,72,73,74]. Our findings tell a different story of when older Canadians were confronted with transitioning back into open spaces, albeit in a grocery store, a bank, or a drugstore. Under the auspices of establishing a ‘new normal’ [32,71] wherein fear and anxiety are commonplace [75], severely anxious participants were less likely to be exercising and getting adequate sleep. Regular exercise seems to provide breathing space and a change of place to dissipate anxiety, at a reasonable distance from others [76]. Inadequate sleep and anxiety can fuel one another [77]. These are back-to-basic, familiar creature comforts. Practitioners and program planners should not take for granted the mental benefits of getting out for a walk [78] or a sleep routine [79]. Simple can be better when one’s health-related stakes are high.

We are far less inclined to recommend information management. Seeking credible information from sources like WHO and Health Canada appeared not to be a statistically significant between-anxiety-group differentiator. Our findings suggest that it may not be unusual for older Canadians finding a balance between staying informed and taking breaks from news to experience lesser and greater odds of experiencing severe anxiety. This might also be the case when they are unplugging from electronic devices. These patterns support others’ contentions that the advent of COVID-19 has fueled people’s appetites for being in the know, and on a variety of news platforms, even if it frightens them [80,81]. That the pros and cons of information seeking remain a topic of public interest and debate [82,83,84] make our findings about managing streams of COVID-19 information an ideal icebreaker for patient–practitioner conversations. Wide confidence intervals in all three instances are telling, calling for cautious optimism for COVID-aware aficionados.

Our findings suggest that severely anxious participants tended to practice relaxation and meditation. Others report negative associations between mindfulness and mental distress, and that meditating and relaxing appear to dampen symptoms of anxiety, depression, and stress [85,86]. Meditation involves acknowledging positive or negative thoughts and without prejudice while attending to muscle tension, the rising and falling of one’s chest, or the sound of subtly spoken words [78]. We wondered whether the very thought of transitioning into open spaces with COVID-19 lingering rendered some participants’ overly in-sync with symptoms like muscle tension or feeling that something terrible was about to happen. Meditators’ odds of experiencing severe anxiety in the larger population could be 4-fold higher than what the observed odds ratio for this strategy suggests. We therefore recommend connecting older people with physiotherapists or occupational therapists to explore older people’s intentions and anticipated mental drawbacks and benefits. Adding soundscapes might be beneficial. Some older Canadians, having been through intensive care experiences, have found these mentally beneficial [87].

In contrast, severely anxious participants appeared to be less likely to decrease other sources of stress in their lives and to perhaps reduce mental clutter. Others’ research findings support enhancing older people’s awareness of thorny sources of anxiety [72,74,88]. Psychosocial stressors like unsafe living arrangements, being isolated from friends, and having unmet health-related needs can take a toll on our anxiety levels [6,89,90]. So too can persistent loneliness [91]. Arguably, COVID-19 magnified these stressors, particularly during lockdowns [92,93]. While older Canadians seemed to be far less likely to experience financial losses and family breakdowns, they were not immune to losing someone close to COVID-19 or to difficulties accessing health care [9]. Losing a significant other is an irreversible loss.

Other research studies conducted in the first year of COVID-19 tell us that mental resilience, particularly during lockdowns, is an invaluable piece of armor. Some resilient older people have seen having an unwelcome viral companion as a means for psychological growth [94] and to a thicker skin to ward off anxiety that relates to pandemic-related stressors like facing precarious health and looking after others when you too need support [95]. Throughout COVID-19, others lacked symptoms of anxiety that fed off each other and felt less alone [96] and had a family who they could, at the very least, talk to or let any friction between them be [97]. Remembered resilience was not a statistically significant between-anxiety-group differentiator in this study. Our findings suggest that participants with severe anxiety seemed to be less likely to remember this.

Resilience reflects how well people bounce back from adversity [98]. Practitioners and program planners should keep in mind that age-as-vulnerable public health messaging early in the COVID-19 pandemic might set the stage for dismissing or downplaying own resilience [99]. The ‘new normal’ age-as-an-asset messaging based on others’ COVID-related research findings could change this. What resilience looks like in older people’s eyes, and how and why it matters, warrants further exploration in practitioner assessments and in mental health program planning. Resilience-related qualitative inquiry can help us better understand the many faces of resilience within the context of a pandemic at future points in time. As perhaps with meditating, remembering that you are resilient could pose time-sensitive benefits.

### 4.3. Implications for Practice and Research

Nationally representative evidence is essential to help pinpoint who is most at risk for severe anxiety and mitigating strategies, ideally lessening or at the very least keeping people afloat [22]. Practitioners and program planners can better anticipate the ‘who’ and ‘how’ of community interventions and initiatives. This is important because mental health seeking is an often-stigmatized practice [24].

Characteristics and strategies of little empirical significance illuminate an all-too-human side to severe anxiety. Our findings contradict others’ claims during COVID-19. Older men, those without chronic illnesses, and those with a life partner are seemingly no longer an anxiety-advantaged demographic when transitioning into open spaces. While education is a well-known determinant of health, neither were older people with a post-secondary education. Perhaps as Bonnano et al. [99] now aver and as Hobfoll et al. [100] have long forewarned, the sudden and chaotic nature of the changes that a pandemic can bring make once-routine tasks into arduous tasks for everyday people. Older Canadians’ dealings with everyday problems in a structured way was somewhat beneficial and somewhat detrimental. Just prior to our study, a national survey revealed evidence of an overwhelming sense of weariness among Canadians of all ages waiting with bated breath for a future carved in cardboard versus stone [101].

Learning who has resided on the unsavory side of anxiety is sobering. Amidst a desire to rebuild our lives and livelihoods, let us not forget that our mental health can change swiftly and unexpectedly [14]. We hope that our findings address the pervasive stigma associated with untoward mental health and speak to the merits of peer-to-peer support. Shared everyday trials and tribulations could take the edge off mental health conversations. Safe spaces and places for older adults to confide can foster empathic understanding and emotional support [14]. Program planners might explore whether older people living with multiple chronic illnesses would be keen to mentor contemporaries encumbered by more severe forms of anxiety. Education for healthcare providers, caregivers, and family members on how to show empathy and support for older adults experiencing mental health issues is an important consideration. Empathy training can enhance the clarity and sincerity of communication and thus foster mutual understanding and better mental health outcomes [66].

Our findings highlight the urgent need for a paradigm shift in how we look at and how we treat pandemic-related anxiety post-COVID-19. It seems rather benign for us to encourage people recovering from life-threatening events to take time to relax and recharge their physical and mental energy [85,86]. The same could be said for stopping ruminating or worrying over things you cannot control [69], and perhaps even to strive to accept your circumstances [78]. Working with older people who gravitate towards strategies that could lessen or heighten their odds of experiencing severe anxiety, albeit remembering resilience or meditating or seeking support, demands an empathic approach and open-mindedness. Cutting down on caffeine or substances, and being kind to yourself, were also tenuous strategies.

Program planners and practitioners should consider integrating more comprehensive and tailored approaches to anxiety management to procure interventions that address the unique challenges faced by older adults. Anxiety management remains a global concern. Severe anxiety disrupts anyone’s ability to tend to their homes, banking and shopping, or caregiving, and none of us are immune to it. Policies are needed to render mental health services as essential as physical health services, and with a mindset towards pandemic preparedness now and for the future. Policies that prioritize public awareness to better engage communities and normalize risk communication are essential (for example, see Hu et al.) [102]. So too are campaigns to dispel myths, reduce stigma, and encourage open discussions about mental health [103]. Mental health campaigns should be inclusive and collaborative endeavors that resonate across cultures [104]. Our findings are meant to spark such conversations.

### 4.4. Limitations

While the findings of this study offer important insights, they should be interpreted with the following limitations in mind. Cross-sectional findings, regardless of how corresponding they are, do not make us soothsayers with respect to anxiety severity [23]. For example, contrary to consistently observed higher levels of anxiety among women during COVID-19, our post-COVID-19 findings suggest that older men’s odds of being severely anxious are similar to older women’s prospects. Some might also argue that anxiety has more than two faces. It is not a two-sided coin. Our penchant for looking to the far right of the GAS-10 variable distribution limits what we can learn. Our nationally representative cross-sections of findings tell of the complexity of mental health recovery work within the context of a pandemic. Our intentions were practical, and our post hoc guidance for practitioners and program planners is limited.

We tell of coping strategies of prior significance posing lesser and greater odds of experiencing severe anxiety, and more commonly than one might think for older Canadians at large. What limits us can also make us more insightful. For example, we had no formal measure of resilience. The Brief Resilience Scale [105] generated many informative findings during COVID-19 (for example, see Beringer et al.; Brinkhof et al.) [60,96]. We used a lay tool that posed a single question to reduce responder burden. Wister et al. [18] tell of differences in mental health betterment with respect to older Canadians’ mental, social, and functional resilience. Disaster management experts tell of resilience measures seldom robustly predicting who will best recover over time [100]. Perhaps expecting the unexpected is the ‘new normal’ post-COVID-19 [23]. In the meantime, we ask researchers to ardently speak to observed strategy-specific confidence intervals and *p*-values.

This study, while nationally representative, may not fully account for cultural differences in the perception and reporting of anxiety, and the use of coping strategies [71]. Canada is home to some 450 ethnic and cultural origins [106]. Research tells us that older Black, Indigenous persons, and people of color have experienced more symptoms of anxiety than Caucasian older people [60]. It was beyond the scope of this study to compare participants by ethnicity. There may be cultural differences in the perception and reporting of anxiety, and the use of coping strategies, which may affect the number of people facing severe anxiety. This limits our ability to contribute to the much-needed discourse about cultural nuances of mental health recovery work, and to further apprise practitioners and program planners accordingly.

## 5. Conclusions

This study uniquely investigates differences in personal characteristics and coping strategies among older Canadians experiencing severe anxiety versus none at all. We shed light on what early recovery work might look like post-COVID19 for a nationally representative sample of Canadians ranging from 60 to 85+ years of age. While our findings highlight significant patterns and suggest practical guidance for recovery efforts, they also underscore the complexity of mental health recovery in this population. Practitioners and program planners should focus on developing more targeted and holistic approaches to meet the distinct needs and challenges of older adults.

## Figures and Tables

**Table 1 healthcare-12-01837-t001:** Personal characteristics and general health status of participants (*n* = 606).

Characteristics	No Anxiety at All (*n* = 290) *n* (%)		Severe Anxiety (*n* = 316) *n* (%)		*X* ^2^	φ	*p*-Value
**Age (years)**							
60–69	143	(49.5)	211	(67.0)	21.29	0.187	<0.001
70–79	101	(34.9)	81	(25.7)			
80 and above	45	(15.6)	23	(7.3)			
**Sex**							
Female	121	(42.6)	196	(63.4)	25.79	0.206	<0.001
Male	163	(57.4)	113	(36.6)			
**Gender Identity**							
Transgender and non-binary	41	(15.4)	45	(15.9)	23.20	0.196	<0.001
Cisgender woman	91	(34.1)	149	(52.7)			
Cisgender man	135	(50.6)	89	(31.4)			
**Number of chronic illnesses**							
None	109	(38.7)	132	(42.6)	1.26	0.045	0.532
One	80	(28.4)	77	(24.8)			
Two or more	93	(33.0)	101	(32.6)			
**Perceived Health**							
Poor/fair	44	(15.3)	156	(51.1)	86.50	0.378	<0.001
Good	143	(49.7)	96	(31.5)			
Very good/excellent	101	(35.1)	53	(17.4)			
**Post-secondary Education**							
No	192	(66.2)	185	(58.7)	3.594	0.077	0.058
Yes	98	(33.8)	130	(41.3)			
**Marital Status**							
With partner	175	(60.3)	174	(55.1)	1.72	0.053	0.189
Without partner	115	(39.7)	142	(44.9)			

Note: A chi-square test (*X*^2^) was used to compare categorical variables between participants with a GAS-10 = 0 score and those with GAS-10 ≥ 12. Frequency totals for some independent variables (i.e., age) do not add up to 100% due to missing data.

**Table 2 healthcare-12-01837-t002:** Logistic regression analysis showing personal characteristics associated with severe anxiety among the participants (*n* = 606).

	Crude		Adjusted #	
Variable	OR (95% CI)	*p*-Value	AOR (95% CI)	*p*-Value
**Age (years)**				
60–69	Ref		Ref	
70–79	0.544 (0.379–0.780)	<0.001 ***	0.437 (0.248–0.772)	0.004 **
80 and above	0.346 (0.201–0.598)	<0.001 ***	0.230 (0.098–0.543)	<0.001 ***
**Sex**				
Female	Ref		Ref	
Male	0.428 (0.308–0.595)	<0.001 ***	0.590 (0.171–2.036)	0.404 ^a^
**Gender Identity**				
Transgender and non-binary	Ref		Ref	
Cisgender woman	1.492 (0.908–2.452)	0.115 ^a^	0.689 (0.267–1.777)	0.441 ^a^
Cisgender man	0.601 (0.364–0.991)	0.046 *	0.657 (0.249–1.734)	0.397 ^a^
**Number of chronic illnesses**				
None	Ref		Ref	
One	0.785 (0.531–1.189)	0.264	0.904 (0.494–1.654)	0.743
Two or more	0.897 (0.614–1.310)	0.573	0.632 (0.353–1.133)	0.123
**Perceived Health**				
Poor/fair	Ref		Ref	
Good	0.189 (0.124–0.289)	<0.001 ***	0.196 (0.110–0.351)	<0.001 ***
Very good/excellent	0.148 (0.092–0.237)	<0.001 ***	0.138 (0.068–0.279)	<0.001 ***
**Post-secondary Education**				
No	Ref		Ref	
Yes	1.377 (0.989–1.3917)	0.058	1.372 (0.811–2.324)	0.239
**Marital Status**				
With partner	Ref		Ref	
Without partner	1.242 (0.899–1.716)	0.189	1.026 (0.620–1.697)	0.920

Abbreviations: OR = odds ratio, AOR = adjusted odds ratio, CI = confidence interval, Ref = Reference. Notes: # The adjusted logistic regression model consisted of age, sex at birth, gender identity, number of chronic illnesses, perceived health, education, and marital status. The goodness-of-fit test for the multivariate logistic regression was the Hosmer–Lemeshow (*p* = 0.282) chi-squared test; *p*-values: *** *p* < 0.001, ** *p* < 0.01, * *p* < 0.05. ^a^ not statistically significant.

**Table 3 healthcare-12-01837-t003:** Multivariate logistic regression analysis showing coping strategies associated with severe anxiety among the participants (*n* = 606).

Variable	AOR (95% CI)	*p*-Value
I accepted that some fear and anxiety was normal	4.76 (2.031–11.200)	<0.001 ***
I sought credible information, i.e., WHO, Health Canada, Provincial Ministry of Health, Local Public Health Unit	0.88 (0.505–1.540)	0.657
I found a balance by staying tuned in (open to new stories about COVID-19) but knowing when to take a breather	1.26 (0.649–2.445)	0.494
I brought an intentional mindset to unplugging from electronic devices, including phones, tablets, and computers	1.04 (0.584–1.882)	0.874
I dealt with problems in a structured way	0.62 (0.310–1.249)	0.182
I remembered that I am resilient and was careful with the WHAT-IFs (asking ‘what if’ questions)	0.54 (0.267–1.092)	0.086
I challenged worries and anxious thoughts	5.21 (2.663–10.228)	<0.001 ***
I decreased other sources of stress in my life	0.39 (0.211–0.729)	0.003 **
I practiced relaxation and meditation	2.36 (1.365–4.112)	0.002 **
I sought support from loved ones	1.69 (0.979–2.945)	0.060
I was kind to myself	0.70 (0.310–1.580)	0.391
I ate healthy	0.72 (0.367–1.445)	0.364
I avoided substance use, including smoking, vaping, and alcohol	1.57 (0.899–2.775)	0.112
I had a moderate caffeine intake	1.34 (0.738–2.442)	0.335
I got proper rest and sleep	0.52 (0.275–0.995)	0.048 *
I stayed active	0.21 (0.110–0.408)	<0.001 ***

Abbreviations: OR = odds ratio, AOR = adjusted odds ratio, CI = confidence interval, Ref = Reference. Notes: The adjusted logistic regression model consisted of age, sex at birth, gender identity, number of chronic illnesses, perceived health, education, and marital status. The goodness-of-fit test for the multivariate logistic regression analysis was the Hosmer–Lemeshow (*p* = 0.282) chi-squared test; *p*-values: *** *p* < 0.001, ** *p* < 0.01, * *p* < 0.05.

## Data Availability

The dataset generated during and/or analyzed during the current study is available from the corresponding author on reasonable request.

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
