# Peer review of "Factors Associated with Older People’s Anxiety Symptom Positioning after COVID-19: Cross-Sectional Findings from a Canadian Sample"

_healthcare, 2024, doi:10.3390/healthcare12181837_

Round 1

Reviewer 1 Report

Comments and Suggestions for Authors

Dear authors, regarding your article “Storytelling (tails): Factors associated with older people's anxiety symptom positioning after COVID-19”, I would like to address a few comments: 

In the abstract, where the text says “They were not significantly more likely to be women”, I suggest instead “There were no significant differences bewtween sexes”.

In the introduction there is a gap in the text in the word “loneliness”.

Where the text says “Older people take advantage of years of life experience to build a cadre of resources and strategies to improve their mental health”, references 7 and 8 are provided, however 8 is a guideline that does not provide any support for this statement, and 7 is an Australian study that does not make such a claim either, the authors specify that older people with coping mechanisms and resilience have better mental health outcomes, not that with increasing age these mechanisms also increase.

In methods, in the sample section, I suggest adding the year in which the data were collected.

In the sample size calculation, I suggest adding the rest of the parameters for replication purposes. 

Contrary to your abstract, Table 1 shows significant differences in the two study groups on the covariates age, sex, gender identity, and perceived health. I suggest that you add an effect size dimension to the differences, such as the Odds Ratio.

Table 3 appears to have the odds ratio backwards, e.g., “I practiced relaxation and meditation” was significantly associated with more severe anxiety with an AOR of 2.36. This analysis probably shows the opposite, that coping mechanisms were associated with NOT having severe anxiety.

First paragraph of the discussion has different spacing than the rest of the paper.

The discussion talks about some non-significant associations as if they were, e.g., finding credible sources of information.

Where it says “The practice of relaxation and meditation was more common among participants with severe anxiety, and this contradicts the findings of others”, I would like you to consider that “correlation does not mean causation”, what your results show is that people with severe anxiety tend to meditate more, probably because of anxiety, but the result does not show that anxiety is worsened or not relieved to some extent by meditation. Their results do not challenge the conclusions of references 84 and 85.

Author Response

Thank you very much for taking the time to review this manuscript. Please find the detailed responses below attached and the corresponding corrections highlighted/in red in the re-submitted files.

Reviewer 2 Report

Comments and Suggestions for Authors

First of all it was an honor to read your work.

The paper it's well build, clear and easy to read. The references are solid and well used. The methods were well chosen and the results well presented with interesting results for further research.

I just have two little suggestions of improval:

1. The title could be, somehow, deceiving because you start with "Telling tales" and this point to knowing stories that, in fact, it's not the case. I suggest that you change this part of the title.

2. Why you defined 60yo as minimum age for entering the study when in Canada WHO define that 65yo it's the beginning of old age. You must refer some references to support this choice.

All else in the paper it's perfect. Good work!

Author Response

Thank you very much for taking the time to review this manuscript. Please find the detailed responses below attached and the corresponding corrections highlighted in red in the re-submitted files.

Reviewer 3 Report

Comments and Suggestions for Authors

The authors present a theme of actuality that is characterized by importance and strong medical interest.   The manuscript is well written and I consider that all main points have been debated.  

The suggestions I have is that:  

English spelling should be revised. In the abstract, the age for the participants and some inclusion criterias, how the study was performed, the period of time the study was performed in material and method and some specific results (data analysis perhaps) can be added

Please rephrase the first paragraph from introduction. It is not very clear. Please point more clear the presented ideas just in one tense time (eg: present, past tense) and perhaps in short sentences

“Loneliness”, please rewrite in one word

“Anxiety characterized by excessive fear and worry and behavioral disturbances… “, maybe better “Anxiety characterized by excessive fear, worry and behavioral disturbances”

At reference 23 in introduction I do not understand the term “transpire” and “well”- what? Please explain

In material and method:

Please add the inclusion and exclusion criterias used in your paper

Please add the license number for using GAS 10 or explain if the Geriatric Anxiety scale is a free scale to apply

The statistical analysis (point 2.3) should be included in the results section , in material and methods you can only remember the statistical program that have been used and what have you searched in statistical terms and point of view

Please include each table after its’ explanation from the text

In discussion section :

“a hospital”, please correct, also “death” not dying…; also I do not understand if you refer to your research or to researchers… please rephrase Implication for practical research can be included in the discussion section

Also, the limitations can be included in the discussion section or as a separate entity but they should not appear mixed with conclusions

Separate paragraph for the main ideas- conclusions- is recommended

Comments on the Quality of English Language
 Extensive editing of English language required.

Author Response

(The authors gave the same response as above.)

Round 2

Reviewer 3 Report

Comments and Suggestions for Authors

All the suggested corrections or additions has been maden by the authors.